# Migrant Organizations and Social Protection in Germany: The Functions of MOs for Their Target Groups' Social Protection Practices

Lisa Bonfert [1,*] , Eva Günzel [2] and Ariana Kellmer [3]

1 Faculty of Social Sciences, Technical University Dortmund, 44227 Dortmund, Germany
2 Chair of Sociology/Urban and Regional Studies, Faculty of Social Science, Ruhr-University Bochum, 44801 Bochum, Germany
3 Institute Work and Qualification, University of Duisburg-Essen, 47057 Duisburg, Germany
* Correspondence: lisa.bonfert@tu-dortmund.de

**Abstract:** This article engages with the functions assumed by migrant organizations (MOs) in Germany in the context of the social protection of people with migration biographies. Based on document analyses and qualitative interviews with three groups of actors, we identify four functions through which MOs contribute to their target groups' social protection practices, and show how diverging perceptions toward these functions shape the current role of MOs in a changing welfare system. In addition to providing social services themselves (service function), they mediate with the welfare system (hinging function) and advocate for the interests of people with migration biographies in public and political discourse (advocacy function). Moreover, we demonstrate that these functions are shaped and complemented by a "homemaking" function, a form of informal protection based on mutual support, trust and understanding. In this article, the discussion of the specific ways in which these functions play a role for the social protection of people with migration biographies is based on joint analysis of three data sets. Thus, we juxtapose the viewpoints of MO representatives, their target groups and people associated with welfare state institutions and political administrations. In this way, we show how MOs use these various functions to actively engage with a changing welfare landscape, whereas welfare institutions and political administrations often perceive of the work undertaken by MOs rather as an 'integration'-oriented prerequisite for their own social service provision. As a result, contrasting and sometimes competing perspectives challenge the role of MOs within the German welfare system, even though these organizations already fulfill key functions for their target groups' social protection.

**Keywords:** migrant organizations; social protection; welfare; migration

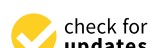

## 1. Introduction

Traditionally, the German welfare state has been characterized by a subsidiarity-based welfare corporatism, in which nonprofit actors (especially the six leading welfare associations[1]) have a privileged position in the provision of social services (Reichenbachs 2018). Since the 1990s, however, the German welfare state system has increasingly evolved toward a pluralizing "welfare market", where established welfare associations face growing competition from a range of private profit and nonprofit actors (Ledoux et al. 2021). In this context, migrant organizations (MOs) have become relevant actors in providing largely voluntarily organized social services targeted especially at the growing number of people with migration biographies (Halm et al. 2020; SVR 2020). Whereas the nonprofit sector has always played a key role within the German welfare architecture (Anheier and Seibel 2001, p. 30; Hallmann 2016), the role of organizations from and for people with migration biographies in this context has long been neglected. Originally referred to as *Ausländervereine* (foreigners' associations), these organizations were seen mainly as spaces for newcomers to

preserve and engage in the languages and traditions of their origin countries (Blätte 2014, p. 13). In recent years, however, discourses around MOs have increasingly focused on the ways in which they may hinder or promote the political goals of 'integration'[2] (Pries and Sezgin 2010, pp. 9–10). As a result, they have garnered growing public and political attention, especially in the course of professionalization and in broadening the scope of their activities (Bonfert et al. 2022; Hunger and Metzger 2011; Schultze and Thränhardt 2013; Halm and Sauer 2015; Halm et al. 2020; SVR 2020; Klie 2022).

Against this background, growing numbers of MOs in Germany began to offer social services targeted specifically at people with migration biographies, especially in the field of education (Halm et al. 2020; SVR 2020). At the same time, established welfare associations and state providers of social services sought to address and adapt to the specific needs and challenges of migrant populations. This is evident, for example, in the concepts of intercultural opening and the provision of specific services (Papen Robredo 2017; Gögercin 2018b, p. 770). Despite these efforts, however, research has shown that newcomers in particular experience certain barriers when it comes to accessing and using formal welfare services (cf. Weiss 2018; Gögercin 2018b; Eurofound 2015). Alternatively, a number of scholars have shown that people with migration biographies draw on a range of protective resources, including not only welfare state services but also informal network-based social protection strategies—as well as migrant organizations (Barglowski and Bonfert 2022; Faist et al. 2015; Faist 2017; Freise and Zimmer 2019; Mumtaz 2021; Saksela-Bergholm 2019; Serra Mingot and Mazzucato 2017).

Despite the surge in MOs' engagement with social services, information is still scarce concerning the perspectives among established welfare providers on the one hand and the welfare targets engaged with MOs on the other when it comes to their roles in the field of social protection. Against this background, we investigate how MOs interact with the welfare system and shape the ways in which people with migration biographies manage social risks. Specifically, we examine the different perspectives of MO representatives, their target groups and actors engaged in established welfare state institutions, associations and politics in order to identify the different functions MOs fulfill for the social protection of people with migration biographies within a transforming welfare landscape, as well as the tensions these differences create across the various actors involved.

Departing from the literature that is concerned with the functions of MOs more generally, we examine previous research on MOs' role within the German welfare architecture. Thereafter, we will present the research design of the collaborative study that underlies this article. Subsequently, we will introduce four functions ascribed to MOs based on the various perspectives we studied and will discuss perceptions about their distinct and entangled contributions to the social protection practices of the people who use the services they offer. Consequently, we will show how MOs engage in social protection strategies within a transforming welfare landscape.

## 2. Migrant Organizations and Their Functions

According to the literature, the term 'nonprofit organizations' (NPOs) broadly refers to "private institutions with a common, public purpose" (Anheier and Seibel 2001, p. 9). In Germany, registered associations in particular have a long-standing tradition that dates back to the Middle Ages (Hallmann 2016, 13f). Today, according to Anheier and Seibel (2001), "the nonprofit sector in Germany is both a basic institution of civil society, with a rich tapestry of associations and organizations of all kinds, and an integral though distinct part of the welfare state and its regulatory framework" (Anheier and Seibel 2001, p. 30). In the course of their historical development, NPOs in Germany thus fulfill a variety of functions that often reach far beyond specified organizational goals.

As recent research has shown, the same holds true for migrant organizations,[3] a term which here refers to the collective organization of people with migration biographies (Fauser 2016, p. 1). In keeping with NPOs' characteristic of mutual interest as a common premise, MOs "are the visible sign of new collective formations and the active engagement of

migrants to improve their situation and follow their interests" (Fauser 2016, p. 2). Although the literature offers no coherent definition of MOs, the study described in this article considers them to be organizations that "allow people with an immigrant background to play a significant role in terms of membership, leadership, and internal structure" (Pries and Sezgin 2012, p. 10).[4] In order to address the manifold challenges and needs experienced by the diverse group of people with migration biographies, MOs rarely limit their activities to just one social and societal function but instead pursue a multidimensional and dynamic approach (Pries 2010, p. 21).

Originally, foreigners' associations (*Ausländervereine*) were perceived as a way for people who had moved to Germany from afar to find a common outlet for exchanging memories and experiences with one another; however, toward the end of the 1980s, the potential functions of these associations became a critical focus of public interest (Blätte 2014, p. 13). Following a period of growing immigration by guest workers without much political consideration, contemporary debates about reforming the immigration law faced public outrage for their nationalist orientation and neglect of the interests and perspectives of these migrant groups (Ibid., pp. 13–14). The protests that followed these debates initiated the wide-ranging mobilization of migrant groups in Germany (Ibid.). In the course of these events, "migrant organizations have become established actors at the level of local politics [ . . . ], have allied as unions and perform as interest organizations at regional and national levels" (translated from Blätte 2014, p. 15). Hence, MOs became key actors in the realm of political representation and were increasingly recognized by the state and the public (Thränhardt 2013, pp. 5–6).

In this context, MOs became more visible within 'integration' debates, while other activities and functions received little attention. On the one hand, the dangers of segregation and extremist tendencies possibly promoted by MOs were perceived to jeopardize the political goals of 'integration' (Pries and Sezgin 2010, pp. 9–10). On the other hand, these actors were also portrayed as contributors and partners in 'integration' politics (Ibid.). Beyond that, however, the civic engagement of people with migration biographies and the organizations they founded and managed in Germany did not receive further attention from the scientific community for a long time, and the structural aspects of these associations were hardly addressed at all (Pries 2010, p. 17; Thränhardt 2013, p. 7). Since the enactment of the Immigration Act 2005 and the National Integration Plan, however, studies concerned with MOs have become increasingly diversified. In this context, scholars have uncovered a growing scope of activities undertaken by MOs that are being increasingly professionalized and diversified (e.g., Koşan 2008, for the city of Dortmund; Hunger and Metzger 2011; Die Beauftragte der Bundesregierung für Migration, Flüchtlinge und Integration 2011; Schultze and Thränhardt 2013; Halm and Sauer 2015; Halm et al. 2020; SVR 2020; Klie 2022).

Therefore, multifunctionality is a key characteristic of many MOs,[5] a finding that has been highlighted in various studies (see, for example, Gaitanides 2003; Pries 2010; Nagel 2016; SVR 2020; Klie 2022). Especially in the context of integration politics, some of the key functions ascribed to MOs include their support and mediating functions, through which they provide assistance in navigating municipal administration systems (Ibid.). Activities that support MOs' target groups to access resources and to connect with people relevant for participating in the education system or labor market, for example, are often framed in terms of "bridging" or "networking" functions (Ibid.). Another set of activities often found in MOs is concerned with political representation of people with migration biographies and awareness-raising for specific challenges they encounter, including racism. These are also referred to as "voice" or "empowerment" functions (Ibid.). Moreover, MOs importantly contribute to the formation of group identities and relationships of trust (Ibid.). In this way, they also engage in activities related to what Boccagni and Vargas-Silva (2021) refer to as "homemaking". Finally, recent studies have shown that many MOs increasingly offer more formalized social services of their own (Halm et al. 2020). Therefore, they play a growing role as "an integral though distinct part of the welfare state and its regulatory framework" (Anheier and Seibel 2001, p. 30). Nevertheless, the specific functions of MOs in the realm

of social protection remain largely unexplored. Therefore, this paper seeks to illuminate the variety of functions fulfilled by MOs for the social protection of people with migration biographies in the German welfare state.

## 3. Migrant Organizations and the Welfare State

Historically, the nonprofit sector has contributed greatly to the formation of the German welfare state architecture. In the wake of the concept of "privileged partnerships" between the state and NPOs, welfare associations were able to establish and professionalize a variety of social services (Hallmann 2016, p. 16). In accordance with the premise of an activating welfare state (Lessenich 2012), this way of pursuing political interests while simultaneously promoting social participation brought forth a range of services offered by welfare associations, including hospitals, nursing homes, youth facilities and daycare centers (Hallmann 2016; Karstein 2013). Today, the welfare landscape in Germany is still strongly influenced by the historically grown corporatist relationship between the state and the Free Welfare Associations. In the sense of subsidiarity, the state assigns social protection tasks to independent institutions, which serve the state as trusted partners and as experts in the field of social protection (Anheier and Seibel 2001, 96ff).

Because their central goal is to secure people's well-being in Germany, welfare associations now include migrant populations as a target group since the beginning of the so-called guest worker migration. Consequently, these organizations also began to see themselves as advocates for people with migration biographies and their concerns (Gögercin 2018a). At the same time, however, numerous MOs are increasingly offering social protection services to migrant groups (Halm et al. 2020; Hoesch and Harbig 2019). In their study of welfare services that are offered by secular MOs in Germany, Halm and Sauer (2015) and Halm et al. (2020) depict an institutionalizing field of migrant welfare in Germany whereby MOs represent migrants' interests and provide social services to diverse target groups. Through a bottom-up, largely voluntary process, MOs offer a broad spectrum of selected services that cover the areas of social affairs, education and culture (Halm et al. 2020, 118ff). Besides formal offers, informal structures increasingly contribute toward enabling people to manage social risks, especially in MOs that provide refugee assistance (Ibid., p. 105). While presenting themselves as new actors when it comes to providing social protection, MOs also function as contact points and advisors for welfare associations in migrant-related matters (Ibid., p. 46).

These recent findings demonstrate that MOs are increasingly engaged in the provision of social services in diverse ways (Halm et al. 2020; Hoesch and Harbig 2019; SVR 2020) and thereby actively contribute to a changing welfare landscape. In this article, we go beyond the classic notion of social protection—that is, as practices designed to manage economic and physiological risks (including unemployment, poverty and occupational disabilities, as well as illness, aging, family burdens etc.)—in order to depict the variety of practices that are potentially involved in social protection, especially in the context of migration. Rather, we employ a broader understanding of social protection that encompasses migrants' personal needs for safety, belonging and self-efficacy as potential contributors to social protection strategies (Honneth 2003; Sainsbury 2006). Therefore, we take into account the range of individual social risk–averting strategies employed by people to secure their well-being within and across national state borders, including both formal welfare services and network-based forms of social protection (Bilecen 2021; Bilecen and Barglowski 2015; Boccagni 2017; Faist et al. 2015; Mumtaz 2021; Saksela-Bergholm 2019).

Although MOs have already begun to play an increasingly important role when it comes to how people with migration biographies manage social risks, the organizational element of social protection is rarely addressed in research on migration-related social protection. Against this backdrop, we analyze the various functions that MOs fulfill for the social protection of people with migration biographies in the context of a transforming welfare landscape in Germany. Specifically, we will demonstrate the interplay of three different points of view. In addition to illuminating the ways people with migration biographies

organize their own social protection and the role they ascribe to MOs (micro-level), we will also illustrate how MO representatives perceive their own role in the provision of social welfare (meso-level). Finally, we also include the perspectives of formal welfare state actors and political administrations toward the role of MOs (macro-level). Drawing on document analyses and interviews with all these groups, this article thus illustrates the complex net of expectations and perceptions toward MOs in the German welfare state.

## 4. Research Design

The analysis presented in this paper is based on a collaborative research project conducted in three cities in the Ruhr region in Germany. With the goal of illuminating the role of MOs within the German welfare architecture and for the people who use their services, three local universities were involved in carrying out document analyses and interviews with various stakeholders between May 2020 and October 2022.

First, 18 semi-structured expert interviews with formal welfare state actors served to explore the ways in which political representatives and the people engaged in the formal provision of social services perceive MOs' embedding in political structures and the local welfare architecture. Second, 15 semi-structured interviews with MO representatives were aimed at investigating the cooperative network structures between MOs and other social service providers. For this purpose, we searched for registered associations, congregations and interest groups of different sizes and with varying scopes of activities that target and are organized by people with migration biographies. Third, 21 interviews and two group discussions with 34 members[6] of 17 MOs allowed us to explore their approaches to organizing social protection and the particular role ascribed to MOs among people who use their services. This group included 17 men and 17 women with a variety of educational and migration backgrounds, occupations, legal statuses and places of origin. While contacts with formal welfare state actors and MO representatives were established through publicly accessible information, the latter group served as important gatekeepers for organizing interviews with people who use the services offered by the MOs. The interviews were conducted in German, and MO representatives offered their support to their members during the interviews when language barriers arose.

In addition, we collected egocentric network charts using Vennmaker[7] (Gamper and Herz 2012; Petermann 2008), in which MO representatives indicated their networks with organizations, institutions and individuals. For the purpose of exploring MO members' own social protection strategies and the particular role ascribed to MOs, we combined questions on participants' biographies and approaches to defining and managing social risks with the egocentric network charts that depicted their personal social protection networks.

While the initial sessions of computer-assisted open coding using MAXQDA (a software program for qualitative research) allowed us to sight and structure the material, theoretical coding and thematic comparisons in accordance with the methods of Meuser and Nagel (2005) provided the basis for identifying key themes that crossed all three working packages. Moreover, the network charts we collected were visually reprocessed (Krempel 2005) and analyzed using applicable structural parameters (Petermann 2005). To learn from the individual social protection practices carried out by MO members, we further employed social scientific hermeneutics, building on in-depth sequential analysis (Amelina 2010; Barglowski et al. 2015).

## 5. The Functions of Migrant Organizations within the German Welfare Architecture

In the context of a transforming welfare landscape, our data show that MOs as specific forms of NPOs take on various functions that contribute to their target groups' social protection practices. In addition to a *service function* (evident in the provision of self-organized services), we also identify *hinging* and *advocacy functions* as key mechanisms for bringing people closer to formal welfare services and communicating their specific needs and challenges to welfare institutions and the wider public. These three functions are

common features of the nonprofit sector, but we found that a fourth function—*homemaking*—plays a distinct role in terms of how MOs approach and manage the social risks that their target groups face. Based on familial relationships and notions of belonging, MOs provide guidance and help these groups become oriented in an unfamiliar and sometimes challenging welfare landscape.

We found that representatives and members of MOs perceive the *homemaking function* as a major contribution to social protection, enabling people with migration biographies to navigate an "activating" welfare state. However, established welfare actors perceive this *homemaking function* to be an 'integration'-related prerequisite for their own social service offerings and believe that MOs fulfill other functions outside the realm of social protection. With these diverging perspectives toward the *homemaking function* of MOs, we identify various tensions and challenges within a complex web of a pluralizing society ("welfare market") in which the role of MOs as providers of welfare is highly contested.

*5.1. Homemaking Function: Providing Guidance and Orientation in an Unfamiliar Environment*

One key function evident across the range of activities through which MOs contribute to their target groups' social protection is what we refer to as their *homemaking function*. In various interviews with both members and representatives of MOs, research participants especially stressed the familial atmosphere evoking feelings of being at home in their MO. This is driven especially by the range of activities designed to provide occasions for gathering and informal modes of exchange often used by entire families (e.g., coffee circles, cooking events and excursions). In the context of this familial and intimate environment, MOs in our sample were frequently associated with "home" or "family", thereby representing important places for people to come together, and to receive guidance and orientation when faced with social risks. Hamid, for example, described the organization *Lomingo e.V.* in the following way[8]:

> *These are my friends. We are a big community, but at Lomingo e.V., we are like a big family. No matter what religion, what nationality, what skin color we have. We are always friendly, we are always together as friends, like a big family. Yes.* (Hamid, age 19, from Afghanistan)

This way of referring to the MO as "home" or "family" was evident not only in various interviews we conducted with members of small and voluntarily organized MOs in particular, but also in our interactions with their representatives.

> *And if you take this aspect into account, then it's not just getting together and drinking coffee and feeling comfortable, but the women then experience/are here in a protected space and framework and that's very important to create trust and to address things that they've basically been dragging around with them all week.* (*Together e.V.*)

This "protected space" provides a respite from the racism and prejudice often faced by people with migration biographies in Germany, and makes it possible for them to address challenging and stressful topics. Consequently, the *homemaking function* is evident in the ways MOs establish relationships of trust that enable their target groups to seek support for managing social risks. When we joined the women at the *Together e.V.* during one of their weekly get-togethers, for example, they started discussing the role of the hijab and the advantages and disadvantages of wearing it in Germany. In the middle of their discussion, one woman said, "These are the things we discuss here, you know. Without prejudice." This shows how informal communion and shared activities can turn MOs into a "second home" where people feel understood and welcome. As registered organizations that target people with the shared experiences of migration, MOs contribute to the formation of a group identity, which further promotes the pronounced belongingness engendered by these organizations (Barglowski and Bonfert 2022; Karstein 2013).

While both members and representatives of MOs in our sample considered the homemaking function to play a key role for approaches to managing social risks, actors from the political sphere and administration recognize related activities mainly for their contribution

to the social and political participation of people with migration biographies in German society. This way of associating MOs with the political goals of 'integration' is evident, for example, in national and local integration plans, which describe these organizations as important actors for overcoming the challenges that can arise in the course of migration and arrival (Weiss 2013, pp. 21–22). Similarly, our interview partners who work for welfare associations and state institutions strongly emphasized the range of 'integration'-related tasks fulfilled by MOs and considered these tasks to be separate from their own and essentially as a prerequisite for organizing social protection. For them, the *homemaking function* of MOs entails an important type of work that lies beyond the scope of their own duties as providers of social services and is an important contributor to the welfare regime, albeit not immediately interfering with it.

> *Migrant organizations, they are more like the whole environment, the trappings, taking care that the flat is properly arranged, that they, I don't know, get to know, you know, how does the tram work? How does this work here, I don't know, public life?* (Employment agency representative)

These institutions see MOs as preparing people to engage successfully with the welfare system and consider these activities as a precedent to and separate from matters of social protection. By contrast, the narratives expressed by MO members and representatives suggest that the *homemaking function* has direct implications for social protection strategies. They view the MOs' role in managing social risks as providing an opportunity to address personal problems and challenges in the context of informal gatherings and relationships (as described above), in many cases incidentally creating venues where they can also discuss matters of social protection. As the relationships developed in the context of MOs become a key part of their members' social networks, the development of social capital thus becomes a protective resource. Within the space of the organization, the MO members we spoke with ascribed particular importance to informal occasions for exchanging information relevant to seeking employment, education or health care. Alexian, a volunteer and member at the *Path e.V.*, whose parents had migrated from Turkey, described this entanglement between networking and addressing issues of social protection in the following way:

> *Coming back to the networking aspect, mentioned earlier. They don't have to be, let's say, top jobs. It's just about, mainly, let's say, regarding the people who came here in the past years, their degrees are sometimes not recognized. That's why they have to work in the low-wage sector. Then they are happy to get anything, basically. And so they ask: Does someone know of someone somewhere, where I can work? Because, it's like this: Most of them, they want to do something, too. They want to work, too. They don't want to sit at home. And then, I would say, good advice is taken well. And in this way, one or the other also found a job.* (Alexian, age 29, born in Germany)

These reflections on what we call the *homemaking function* show how MO-based opportunities for building social capital are important in influencing social protection strategies. As people develop a shared identity and "find solutions together", they build and strengthen a social network, which becomes a key resource for managing social risks. At the same time, members can also use this social capital for engaging with the education system and labor market. Consequently, social capital built within the context of MOs not only entails informal network-based forms of social protection, but also affects more formalized aspects of social protection (Klie 2022, p. 128).

In this way, tasks that might politically be categorized as 'integration' turn out to be not merely a prerequisite but rather an integral part of social risk–averting strategies. In addition to acting as a "transmitting strap of values and perceptions of norms" in a new environment (Karstein 2013, p. 15), the organizational setting further contributes to people's abilities to build and extend social capital based on their engagement with their organization (Ibid.). This cohesiveness of MOs, which addresses the need for both orientation and support, becomes key for the ways in which participants considered their MOs as influencing matters of social protection. Consequently, from the perspective of

MO members and staff, MOs' *homemaking function* adds importantly to individual social protection strategies by speaking to the often unaddressed need of people with migration biographies for a space where they feel understood and "at home" (Boccagni and Vargas-Silva 2021).

### 5.2. Service Function: Migrant Organizations as Providers of Protective Resources

Based on the diverging perspectives regarding the *homemaking function* of MOs in the context of social protection, the various stakeholders we spoke with perceived the other three functions we identified in different ways as well. To begin with, the *service function* of MOs as providers of various kinds of social services (Halm et al. 2020; Hoesch and Harbig 2019) was repeatedly stressed across the range of interviews we conducted. However, perceptions about the role of this function in social protection practices differed strongly. From the perspective of MO members and representatives, the familial atmosphere and relationships of trust developed in the course of *homemaking* allow MOs to create and offer services that are tailored to the specific needs and experiences of their members. Beyond the informal forms of assistance provided by means of loose get-togethers or support through personal networks, this function also includes a range of more formalized services such as languages courses, childcare, advisory services and workshops on different topics. Some of the MO members we spoke with emphasized the impact these services have had on their lives, such as finding new entry points to the labor market or receiving support in finalizing their diplomas or improving their German language skills, to name just a few. One example is Linh, who had migrated to Germany from Hong Kong with her Chinese-Vietnamese parents when she was a child. In addition to stressing the above-described "protected place" she associates with the *Culture and Hope e.V.*, she particularly highlighted a nursing course she had attended a few years prior to the interview. When her daughter was looking for an internship opportunity in 11th grade, Linh suggested consulting the representative of *Culture and Hope e.V.*

> *There was this dementia course and a nursing course. And I sent my children there, and they received consultation. And my daughter has been in the nursing business ever since. At the age of 20, right? She started the apprenticeship. She was in 11th grade when she first did an internship. She worked with the elderly, and she liked it. So she started the apprenticeship. Like her friend, the same thing. She also came here. I never thought she would like this kind of job. She is very, like, fancy pants, you know? She never changed her siblings' diapers, never did anything like that. But she did this job. When she told me about that, I fell off the chair. Seriously. Because it's not like her. But she did it.* (Linh, age 53, from Vietnam)

In addition to helping Linh's daughter find an internship placement in the nursing sector, *Culture and Hope*'s representative also helped her and her friend with their apprenticeship applications. Two years later, Linh herself started working in the same nursing service where her daughter was studying and pursued an apprenticeship in elderly care. In addition to the nursing course, Linh mentioned a variety of courses she had attended and benefited from, including sewing, computer and language courses.

This example shows how the trusted relationship with MOs encourages members to take part in formalized courses and to seek support in the various spheres of life. Consequently, from the perspectives of people involved with MOs, the *homemaking function* significantly affects the ways in which their *service function* contributes to social protection practices, because it allows MOs to tailor their own services to members' needs. Especially among the larger and more professionalized MOs, we witnessed a division into work units that was also evident in welfare associations (e.g., labor and qualification, health and mental well-being, education), in which they link both informal and formal types of support. Others specialize in specific areas of social protection (e.g., education). Our data further show that this combination of formalized services and structures, on the one hand, and informal dynamics, on the other, is evident in both mainly voluntarily organized MOs and structurally funded organizations with a broad staff base. Regardless of the degree of

formality, however, the *homemaking* effect based on the establishment of personal networks and the character of the MO as a "home" or a "safe space" play key roles. The *service function* as a contribution to social protection is therefore closely linked to the *homemaking function*, which is important in distinguishing the logic of MOs from that of service-oriented welfare state actors.

Accordingly, from the perspective of MO representatives we spoke with, the advantage of their *service function* for their target groups' social protection lies in its connection with their *homemaking function*. Through their open and dynamic structure and the blurring of boundaries among members, advice-seekers and supporters, MOs and the people who act in their surroundings create ideal conditions for growing interpersonal networks of informal exchanges of information, social and emotional support and advice in all relevant areas of people's lives. In terms of education as an important aspect of social protection, examples include a spontaneous language practice meet-up or homework supervision that members of the MO provide for each other, as well as a target group–oriented introduction to the German educational system and the accompaniment to the parent–teacher conference at school. Consequently, MOs act according to the bottom-up principle (cf. Halm et al. 2020)—that is, they form ad hoc services in accordance with their members' needs. Based on their largely voluntary structures, these services are often available day and night.

While MO representatives and members thus identify a distinct *service function* in the ways they contribute to social protection practices, institutional welfare representatives perceive of the activities pursued by MOs as essentially informal and thus detached from the formal welfare services offered by state institutions. Therefore, they also expressed their doubts regarding the role of MOs as social service providers:

> *I have to admit quite honestly, in the depth of how we understand welfare, I mean, according to my observation, all six associations in NRW. The Jewish community was included as the smallest. Have we, i.e., at least so far in depth, not encountered an MSO that really offers its own services, such as senior services, daycare centers or even structural consultations? Of course, in the case of the ‚Paritätische', they are made up of many small associations and also of MSOs.* (Welfare association Y)

Although this employee of a welfare association acknowledges that MOs offer social services, he doubts their degree of professionalism and structural anchoring. While many MOs we spoke with perceive themselves as independent and often increasingly professional actors in the field, the fact that many of them are members of "Der Paritätische" causes other welfare actors to associate MOs' social protection–related services with this umbrella organization. This finding demonstrates that, in the context of historically grown welfare structures, established welfare associations recognize MOs as hardly relevant players in the welfare system. In addition to ascribing to MOs a lack of professionalism, welfare associations especially criticize the MOs' targeted focus on certain social groups.

> *I think it's difficult to make an ideological decision or a top-down decision. Instead, you have to look at what the need is and who can best meet that need. And if it is a migrant organization, then it should, in my opinion, be awarded the contract. But if it's a migrant organization that is exclusively concerned with the needs of Sardinian women although the need is actually there for all women, then I don't think it should get it.* (Welfare association X)

This statement highlights the market logic of the welfare system, while also stressing the welfare association's mandate to reach all people in need of support rather than focusing on certain target groups. Across the range of interviews with state representatives and welfare associations, we repeatedly found this emphasis on the claim for universalism. Thus, they accuse MOs of a narrow focus on certain clientele and question that the needs of people with migration biographies actually require an alternative or parallel social protection system. Instead, they see their mandate as integrating migrants into existing systems. By contrast, MO representatives highlighted the value of being closely connected to the people they work with in order to provide needs-based services that speak to

individual challenges. The fact that a range of MO members we spoke with emphasized that the social services they offer had significantly influenced their social protection strategies indicates that this particular mix of *homemaking* and *service functions* fulfilled by MOs contributes to the social protection of their members in an essential way.

*5.3. Hinging Function: How Migrant Organizations Ease Access to Welfare*

In addition to providing services of their own, we also found MOs to fulfill an important function in helping their target groups access existing welfare state services. In response to a range of barriers to access shaped by the high demands of an activating welfare state paradigm (Lessenich 2012) and exclusionary logics of welfare states (Carmel and Sojka 2021), MOs play a key role in "building bridges" to existing welfare structures. This *hinging function* is usually discussed in reference to the destination society of migration more generally—not without criticism, however, because it suggests a need to connect two separate groups (Schultze and Thränhardt 2013). In the context of this article, the image of bridge-building as a function related to social protection points to the ways in which MOs enable access to welfare. However, the various groups of interview partners in our study differed in their perceptions about the concrete role of this *hinging function* with regard to social protection practices for people with migration biographies.

The importance ascribed to MOs and their dense networks (Günzel et al. Forthcoming) as entry points to the welfare system was particularly evident in interviews with MO members who described the challenges of seeking social support in formal institutional settings. Samia, for example, came from Morocco at the age of 28 (8 years prior to the interview). Having acquired German language skills, she was eager to become a preschool teacher and obtain the necessary qualifications. In search of support from the local Jobcenter, however, she experienced rejection and discouragement:

> *I asked the Jobcenter if they can help me with an apprenticeship. And she said, no, unfortunately we can't. You have to find something on your own and then we can see what we can do. [ . . . ] And when I say I would like to do an apprenticeship, they say that I have to find a job, go and work. But when you ask for their support, they say no.*
> (Samia, age 36, from Morocco)

This example illustrates the logic of an "activating" welfare state, in which citizens are expected to "find something on [their] own" for securing their livelihood and well-being. However, in seeking help in finding an apprenticeship placement, Samia experienced rejection and resentment. Therefore, she requested support from *Together e.V*. Being familiar with the institutional landscape, the organization's representative began to push for a solution. She helped Samia contact the authorities and find answers as to why her previous diplomas were not recognized. In this example, the hinge to welfare state structures established by *Together e.V*. fills the gap between Samia's needs, on the one hand, and the services offered by the institution for job orientation (the Jobcenter), on the other. Especially because of her trusted relationship with *Together e.V.'s* representative, Samia is able to communicate her difficulties and subsequently rely on this MO to get the help she needs.

These challenges in dealing with the welfare system were also stressed by the MO representatives we spoke with. In order to support their members, many MOs maintain regular contact with these authorities and offer various forms of support. In addition to providing assistance with translations and the overall bureaucratic processes involved, they also accompany their members to official appointments. This self-perception of MOs as hinges between their members and state institutions was particularly evident among voluntary representatives employed at these institutions themselves who engage with this *hinging function* on a regular basis. In these cases, they act as experts who represent the concerns of people with migration biographies while simultaneously distributing the relevant institutional knowledge of German public services. As with their *service function*, MOs' abilities to provide support in this way and the likelihood that people will use it is strongly linked to their *homemaking function*. Based on established relationships of trust,

MO staff are able to learn about the specific challenges their members face and can respond accordingly.

One example of this *hinging function* in the specific context of securing health care was described by a representative of the *Enjo-Suru e.V.*, an organization for people with a Japanese background. She told us about the case of an elderly woman she became aware of while working as an interpreter in the hospital. When she learned that the woman did not have health insurance, she began to mediate between various care centers, nursing homes, health insurance companies and the woman herself, who has been supported by this association ever since. According to this representative,

> *That is, so this is such a part, which I would and could take over voluntarily with my association. Yes, the caregiver has somehow put everything in order, as far as debts are concerned, then also applied for care, first of all a health insurance. So, the lady had apparently not been covered by a health insurance in Germany for more than ten years, so also got sick status again and then applied for care and now also nursing service is there twice a day. This is all such a part of the caregiver and I am now really only culturally native language support there, but I mediate of course between nursing service and this lady, between caregiver and this lady and so in this form.* (Representative from *Enjo-Suru e.V.*)

Because of the various challenges migrants confront in dealing with the health care system in Germany, the support provided by *Enjo-Suru e.V.* was an important turning point for this woman. Especially in the face of language barriers and a lack of information regarding options for receiving health care support, MOs fulfill an important function in helping people with such administrative challenges.

In addition to MOs' ability to establish hinges to the welfare system for their members, we also found evidence of the reverse—cases in which MOs facilitate access to people with migration biographies for welfare associations. This is evident, for example, in their activities as interpreters and, in some cases, as providers of "intercultural training" for employees at the Jobcenter. Moreover, MOs often act as brokers who help employees at administrations or welfare associations to establish contact with clients and to draw attention to their particular services. This relationship shows the ways in which MOs' *homemaking function* contributes to distributing their services to people with migration biographies more easily and also benefits established welfare actors in seeking to reach this group of people.

Although this bidirectional engagement between members of MOs and the institutional landscape may encourage approximation between the two, it also confronts MOs with tension and conflicts of interest. One example from our interviews is the case of the *Ameren e.V.* This MO received a mandate from the local youth welfare office to support a family with critically ill children but was confronted with the family's reluctance to deal with this state actor. *Ameren e.V.*'s representative describes the tension that arose from the MO's *hinging function*, as follows:

> *One is the representative of the Youth Welfare Office in the family, but one also plays the role of the family's advocate before the Youth Welfare Office. [...] So you have such a (hinge function?) and that is not always easy to manage that both sides, both the family that you care for, as well as the youth welfare office, which somehow writes help plans, plans and now sets educational goals, is satisfied, yes. And if other players then come on board.* (Representative from *Ameren e.V.*)

Taking this position as a mediator between migrants and the state forces MOs to negotiate contradictory interests and reveals how MOs see themselves and their offerings as important "bridge builders" between people with and without migration biographies, as well as between their target groups and state authorities. In this role, many MOs in our sample expressed their wish for greater recognition of their work and subsequent funding and political participation. While MOs thus envision more collaboration with welfare state actors as equal partners, our interview respondents from municipal administrations and

welfare associations emphasized that they see MOs mainly as supporters of their own role as social service providers:

> *So if some customer doesn't get what he wants, then he runs to a migrant organization and they get in touch and say, yes, we have this and this case, is there a possibility? That was in the time when—yes, so 2015, 2016 it was very bad, because, there were actually a great many who wanted to help and wanted to help without any legal basis and were of the opinion that they were the ones who ate the knowledge with big spoons. And then there were some problems.* (Employment agency representative)

From this representative's point of view, interventions by MOs, which reach too far into their area of expertise, suffer from a lack of "legal basis" and professionalism. Therefore, they and other welfare state actors in our sample considered cooperation with MOs largely limited to the latter's role as 'brokers' facilitating access for people with migration biographies. This further demonstrates the boundaries drawn by established welfare actors between themselves as 'experts' of social protection, on the one hand, and MOs as essentially providers of orientation and informal support and disconnected from the field of social protection, on the other. At the same time, narratives of MO members and representatives show how social capital established within MOs in the course of their *homemaking function* facilitates access to the information and skills relevant for navigating the welfare system. The tensions and conflicts of interest created by these diverging points of view highlight the challenges faced by MOs that 'interfere' with an established welfare system.

*5.4. Function of Advocacy and (Political) Representation: How Migrant Organizations Advocate for the Specific Needs of People with Migration Biographies in Managing Social Risks*

Representation of the specific needs of people with migration biographies and advocacy for their rights are widely acknowledged as important functions fulfilled by MOs (Blätte 2014, p. 15; Thränhardt 2013, pp. 5–6). Accordingly, their *advocacy function* was frequently stressed in interviews with actors in politics, administration and welfare associations. Our data show that the overall claim of MOs to support their members and to make their demands heard also plays an important role regarding matters of social protection.

Activities in the realm of advocacy and representation may include MOs' participation in (political) committees and lobbying work. Recently, various MOs have begun to advocate for the "cultural opening" of welfare care or regular financing for their organizations, for example. This way of engaging with the political landscape has turned MOs into important contact partners for municipal (integration) administrations, which regularly invite MOs to exchange ideas in order to discuss current needs, tasks and challenges. In the words of one city administration employee,

> *But what I want to say is not only that they [MOs] are recipients of funding, but these are really organizations that develop urban society by also advancing migrant organizations, especially [MO X]. And in this respect, I think as a city employee you also have to humbly stand in front of this achievement and say, okay, this is not a hierarchical thing, but these are people who are really trying to develop [city Z] just like you, like me, to bring [city Z] forward in the field of integration.* (Employee of the city administration)

From this administrative employee's point of view, the ideas and impulses provided by MOs importantly contribute to shaping the city in inclusive ways, because they actively articulate the interests, needs and demands of people with migration biographies. For politicians, this *advocacy function* of MOs is an important way for them to get to know these particular needs in order to incorporate them into their political decision-making.

Because MOs represent the particular concerns of people with migration biographies, they contribute significantly when it comes to enforcing projects and making political demands. As experts in migration and 'integration'-related issues, some of the organizations we interviewed are regularly invited to join discussions at the local and even on the national level with political administrative representatives. Such opportunities, created by MOs

being actively engaged in (political) representation and advocacy, were also stressed in interviews with MO representatives:

> *Yes, first of all, of course, we also try to make people aware of these problems. Even if it's not on a regular basis, we publish a newspaper every two or three months. That's where we address these problems, right? And of course we try to get people to organize themselves, to put more pressure on politicians, so that these problems also get attention.* (Representative from *Ortak e.V.*)

Here, the representative from *Ortak e.V.* described how raising awareness for unequal educational opportunities and poverty may contribute to political responses, especially by publicly communicating the specific problems and challenges faced by the people they work with. Consequently, their knowledge of these problems is an essential prerequisite for adequately engaging in advocacy and (political) representation. In this way, the potential of their advocacy function is also strongly related to their *homemaking function*, which contributes to their members' openness regarding the challenges they face.

Similarly, the representative from *Dorado International e.V.* highlighted the opportunities of advocacy-related activities based on internal knowledge for "sensitiz[ing] decision-makers for the concerns of people with a history of migration". By engaging in political representation and advocacy, MOs were found to fulfill an important role in actively shaping policy-making toward the greater inclusion of people with migration biographies in ways that match their specific needs.

This *Dorado International e.V.* representative wants her MO to be "not just a member" but to have an impact and to actively contribute to the decision-making processes, which is one striking example of how MOs already seek to take part in shaping societal developments. In providing a range of largely formalized services that are relevant for social protection, with organizational structures similar to those of established welfare agencies, the potential of advocacy-related activities for promoting people's well-being and social protection was emphasized:

> *It's not about party politics, it's about how can I improve people's lives on a small scale locally, how can we make life better locally in [three cities in North Rhine–Westphalia], wherever, very practically, no, with an advisory service, with a great event, with a meeting place, with whatever. And how can we improve life, the diverse, pluralistic, democratic coexistence, by improving the decisions, by influencing them. So that we plow that at all levels. And in this respect, of course, we do it at the municipal level, at the state level, at the federal level. If someone asks us, we'll also do it at the EU level, my God. (laughs) Right? So now that's a resource question, what you can do and create. But, exactly. Wherever we can get involved, we try to get involved.* (Representative from *Dorado International e.V.*)

This testimony demonstrates that MOs actively seek to contribute to societal change in ways that consider and include the particular needs and challenges of people with migration biographies who are often unheard and underrepresented in welfare politics. The *homemaking function* of MOs enables them to establish close relationships with the people whose interests they seek to represent, affecting their knowledge as well as their motivation for doing so in important ways. Our data indicate that MOs thus already actively engage in shaping the overall welfare landscape in Germany, even though their contribution to social protection remains largely underrated.

## 6. Conclusions

This paper has demonstrated the various perspectives of welfare state actors and representatives, as well as of MOs' target groups, toward the functions that MOs fulfill in the context of social protection practices among people with migration biographies within the German welfare landscape. Based on document analyses and interviews with various stakeholders, we identified four functions ascribed to MOs, namely *homemaking*, *service*, *hinging* and *advocacy functions*. Specifically, we showed that diverging perceptions of the

role of these functions in matters of social protection create particular challenges for MOs engaged in an established yet changing welfare system.

For representatives and members of MOs, these four functions represent key contributions to their target groups' social protection practices. To some extent, these functions interact and influence each other. In this context, we found the *homemaking function* to play a particularly important role both as a distinct source of social protection and as a prerequisite for the other functions in order to fulfill their purpose. In addition to building networks that serve as informal protective resources, the close relationships established within MOs thus also provide the basis for needs-oriented services (*service function*) and the articulation of their target groups' interests toward welfare actors and politicians (*hinging* and *advocacy function*). In the specific context of MOs and social protection of migrant groups, it is their *homemaking function* that importantly affects how their *hinging*, *service* and *advocacy functions* contribute to their target groups' social protection practices. Here, perspectives of MO representatives strongly resemble those of MO members, which are often difficult to disentangle. In many cases, members become actively engaged as staff, thereby not only benefiting from social protection-related functions of their MOs themselves, but also offering assistance to other members. Based on these shared experiences of receiving and providing support in various ways, their perspectives toward the role of MOs in the social protection of people with migration biographies are very similar.

However, politicians and actors in established welfare agencies and associations have a different perspective regarding their perceptions of these functions. For them, their own mandate to secure people's livelihoods and well-being is clearly separate from the informal tasks they associate with MOs. In distinguishing between formal and informal activities in this way, they describe MOs as voluntary assistants who are important for migrants' orientation in society and in the welfare landscape and who can establish access to their own services. Despite a shift toward an "activating" welfare state with pluralizing welfare actors and expectations toward greater agency in social protection practices in an increasingly diversifying society (Klammer et al. 2017; Ledoux et al. 2021), our findings suggest that established welfare actors advocate for their specific position on the 'welfare market' and thus entertain the work of MOs with reluctance. This stance poses a variety of difficulties for MOs when it comes to competing and consolidating as welfare actors as they confront conflicts of interest and struggles for funding (Kellmer et al. 2022). Nevertheless, the social protection–related functions identified in our study demonstrate that MOs are already an integral part of the welfare landscape in Germany (Anheier and Seibel 2001, p. 30; Bonfert et al. 2022; Halm et al. 2020; SVR 2020). With the range of functions they fulfill in the realm of social protection, MOs not only create access to the formal welfare system but also cover a variety of needs that are not taken into account in the established system. Therefore, they are likely to continue their claims for greater recognition and participation within a shifting welfare architecture.

**Author Contributions:** Conceptualization, L.B., E.G. and A.K.; Data curation, L.B., E.G. and A.K.; Formal analysis, L.B., E.G. and A.K.; Investigation, L.B., E.G. and A.K.; Software, L.B., E.G. and A.K.; Visualization L.B., E.G. and A.K.; Writing—original draft, L.B., E.G. and A.K.; Writing—review & editing, L.B., E.G. and A.K. All authors have read and agreed to the published version of the manuscript.

**Funding:** This research was funded by the Mercator Research Center Ruhr (MERCUR) (Grant number Pr-2019-0049).

**Institutional Review Board Statement:** The research study entitled "Migrant organizations and the co-production of social protection" underlying this article includes human research participants. The study was prospectively approved by the legal offices of the Technical University of Dortmund, the Ruhr-University Bochum and the University Duisburg-Essen. Ethical approval was not mandatory for this study.

**Informed Consent Statement:** Informed consent was obtained from all subjects involved in the study.

**Data Availability Statement:** To protect the privacy of our research participants, research data are not shared.

**Acknowledgments:** We want to thank all the individuals who supported us in our research and who shared their perspectives and personal stories with us. In addition, we are very grateful for all the support and feedback provided by our research team colleagues. Specifically, we want to thank our supervisors Karolina Barglowski, Ute Klammer, Sören Petermann, Ludger Pries and Thorsten Schlee. As project initiators and leaders, they have been responsible for funding acquisition, methodology, project administration and validation.

**Conflicts of Interest:** The authors declare no conflict of interest.

## Notes

[1] The six major welfare associations in Germany include the AWO (Workers' Welfare), Diakonie (a Protestant association), Caritas (a Catholic welfare association), Paritätischer, DRK (the German Red Cross) and ZWST (a Jewish welfare association).

[2] The political concept of 'integration' is highly contested and far from straightforward. Since it often serves as a point of departure for describing the roles assigned to migrant organizations in the literature and in public and political discourse, we use the term here in quotation marks to indicate a political concept targeted toward incorporating people with migration biographies within their destination context.

[3] Migrant organizations (MOs) are also frequently referred to as migrant self-organizations (MSOs). In this article, we use the term "MO" unless the research participants we cite used "MSO" in their interviews.

[4] Essentially based on the characteristic of "migrantness", this definition is highly controversial and all but straightforward. In our search for a common ground for this study, however, we use the term "migrant organization" also based on the self-representation of organizations that took part in this study as organizations from and for people with migration biographies.

[5] In Germany, most of the secular and religious MOs are legally organized as registered associations (*eingetragene Vereine [e.V.]*).

[6] The term "membership" does not refer to any official status but is here used in place of "client", since we found that MOs perceive and refer to the people they work with as "members".

[7] While the network maps contributed important insights on the role of MOs for social protection discussed in this article, their visualizations are not part of this discussion and therefore not presented here.

[8] To protect their identities, the names of all interviewees referred to hereinafter were changed to pseudonyms.

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
