# Peer review of "Migrant Organizations and Social Protection in Germany: The Functions of MOs for Their Target Groups’ Social Protection Practices"

_socsci, doi:10.3390/socsci11120576_

Round 1

Reviewer 1 Report

The author(s) show the diverging perspectives towards the functions of migrant organizations in Germany. The document is well-structured and well-written. Nevertheless, the Abstract is a little confusing.  It is not clear what the research is going on. The following sections provide an excellent overview of the relation between migration organizations and welfare state in Germany. 

Research Design. "we collected ego-centric network charts using Vennmaker", but, where these graphs are represented? There is not a chart in the paper. You don't explain what MAXQDA is. For a non-expert reader in this area, it is difficult to understand what you mean.

Conclusions show as MO give the opportunity to migrant to access to formal welfare but also provide other services useful for those who arrive in the country.

Minor comments:

- Title looks so long, maybe it would be a good idea to change it. 

- Second line of the Introduction, what are these 6 organizations? (AWO, caritas, etc.?) 

Author Response

Dear reviewer,

Many thanks for taking the time to read and comment on our  manuscript. Your feedback was very helpful in making important improvements. You can find our responses to your inquiries in italics below.

Comments and Suggestionsn for Authors

The author(s) show the diverging perspectives towards the functions of migrant organizations in Germany. The document is well-structured and well-written. Nevertheless, the Abstract is a little confusing.  It is not clear what the research is going on. The following sections provide an excellent overview of the relation between migration organizations and welfare state in Germany.

Thank you for pointing this out – we tried to make clearer what the article is about and re-worked the abstract accordingly.

Research Design. "we collected ego-centric network charts using Vennmaker", but, where these graphs are represented? There is not a chart in the paper. You don't explain what MAXQDA is. For a non-expert reader in this area, it is difficult to understand what you mean.

Yes, this was confusing – thank you for mentioning this. We added a footnote stating that, although the article benefitted from the insights we gained from collecting eco-centric network charts, their visualizations were not included in this article specifically. Moreover, we added a short note on MAXQDA, which is a computer software for working with qualitative data.

- Title looks so long, maybe it would be a good idea to change it.

Yes, we agree – and shortened the title accordingly.

- Second line of the Introduction, what are these 6 organizations? (AWO, caritas, etc.?)

Thank you for pointing this out, we added a footnote clarifying what these are.

Reviewer 2 Report

Very well done research: coherent narrative, correct structure and clarity. The references are relevant to the research. And the empirical research is crucial in this paper. In my opinion the conclusion should be more extensive.

Author Response

Dear reviewer,

Thank you very much for your appreciative feedback on our manuscript! You had one comment regarding the conclusion, which you considered to need some extension. Unfortunately, though, we were unsure what exactly you meant by saying "the conclusion should be more extensive". We added a few things where we considered information to be missing. If you have more specific suggestions for us what aspects you would like to be more elaborated, we wpuld be happy to hear them.

Many thanks and best wishes

Round 2

Reviewer 1 Report

The authors followed all suggestions from the previous review. Now, the paper is more readable.

Minor comment:

- Günzel et al. forthcoming. (?)